# A Novel Ethanolic Two-Phase System Based on Deep Eutectic Solvents and Amphiphilic Copolymer for the Extraction of Neohesperidin and Naringin from the Pomelo Peel

**DOI:** 10.3390/foods11172590

**Published:** 2022-08-26

**Authors:** Shanshan Wang, Zongkui Qin, Yicong Wang, Leilei Liu, Zhijian Tan

**Affiliations:** 1Key Laboratory of Hunan Forest Products and Chemical Industry Engineering, College of Chemistry and Chemical Engineering, Jishou University, Zhangjiajie 427000, China; 2Institute of Bast Fiber Crops and Center of Southern Economic Crops, Chinese Academy of Agricultural Sciences, Changsha 410205, China

**Keywords:** ethanolic two-phase system, deep eutectic solvents, extraction, separation, neohesperidin, naringin, pomelo peel

## Abstract

A novel ethanolic two-phase system (ETPS) composed of Pluronic^®^L-64 (PL 64) and deep eutectic solvents (DESs) was constructed for the separation of two similar flavonoids of naringin (Nar) and neohesperidin (Neo) from the pomelo peel. The selectivity (*S*) data showed that DES prepared from tetrabutylammonium bromide (N_4444_Br) and glycerol (Gly) had the optimal distribution efficiency for Nar and Neo standards. A binodal curve of N_4444_Br-Gly/PL 64/ethanol system fitted to the nonlinear Merchuk relationship showed that the biphasic region was narrow for ETPS. The influences of the mass ratio of DESs and PL 64, DES concentration, PL 64 concentration, molar ratio of DESs, temperature, phosphate buffer solution, and ethanolic pH were studied in single-factor experiments. Under the optimal conditions, the maximum extraction efficiency (*E*_Nar_ = 68.32%, *E*_Neo_ = 86.09%), partition coefficient (*K*_Nar_ = 6.66, *K*_Neo_ = 19.13), and *S* (2.88) were obtained in the DESs-rich (bottom) phase. N_4444_Br-Gly, Nar, and Neo with recovery yields of 78.12%, 66.61%, and 68.03%, respectively, had been recovered using D101 macroporous resin. This proposed ETPS is efficient and environmentally friendly and is expected to avail meaningful references for the separation of natural products with similar structures.

## 1. Introduction

As the largest productive fruit in the world, citrus is rich in flavonoids, which is one of the important sources of polyphenols for human intake. Citrus flavonoids (CFs) are the typical representative of polyphenols, which have received much attention in experimental studies due to their multiple pharmacological activities of antioxidant, anti-inflammatory, anticancer, and cardiovascular protection activities [1]. Natural CFs are mainly present in the glycoside or aglycone forms; for example, their typical glycoside structures, namely, neohesperidin (Neo) and narisngin (Nar) (Figure 1), mostly exist in young pomelo fruits and *Aurantii fructus*. Neo exhibits biological activities, including free radical scavenging [2], anti-aging [3], and neuroprotective effect [4], while Nar has pharmacological effects on antioxidant, anti-inflammatory [5], and anticancer [6]. Nar and Neo are used to evaluate the quality of *A**. fructus*, and the Chinese Pharmacopoeia specifies that their contents in *A**. fructus* shall not be less than 4.00% and 3.00%, respectively. Moreover, Nar is also a bitter flavoring agent and can be used as a raw material to synthesize Neo by aldol condensation reaction under the action of an alkaline catalyst [7]. Therefore, it is of great significance to obtain natural high-purity Nar and Neo from citrus.

Due to the poor solubility of CFs in water and organic solvents, as well as the complexity of biological sources, it is challenging to separate and purify natural CFs. Several strategies of ultrasound-assisted extraction [8], matrix solid-phase dispersion [9], and supercritical CO_2_ extraction [10] have been applied to separate CFs. Additionally, a method of D101 microporous resin adsorption coupled with high-speed counter-current chromatography was used to obtain high-pure Nar and Neo [11]. Generally, highly purified commercial products of Nar and Neo can be prepared using an alkaline extraction-acid precipitation method due to their large solubility differences in acid/based solution. Moreover, these methods have defects of time-consuming, biological toxicity, serious solvent loss, and poor separation selectivity [12]. Deep eutectic solvents (DESs), considered green solvents, can be prepared by heating and stirring with a certain molar ratio of hydrogen bond donors (HBDs) and hydrogen bond acceptors (HBAs) [13]. Choline chloride, betaine, and quaternary ammonium salt are frequently used as HBAs and can prepare hydrophilic DESs with HBDs such as polyols, carbohydrates, and carboxylic acids [14]. These DESs have fascinating properties comparable to ionic liquids, including low cost, less-biotoxicity, designability, and suitable biodegradability [14,15]. Moreover, they have the potential to replace organic solvents for the solid-liquid extraction of polyphenols from rosemary [16], alkaloids from herbal medicines [17], polysaccharides from *Ganoderma lucidum* [18], and artemisinin from *Aretemisia annua* [19]. Moreover, due to their designability, DESs had been widely used as green media to construct various liquid–liquid extraction systems for the separation of natural products, such as aqueous two-phase systems (ATPSs) [20], temperature-responsive two-phase systems [21], and dispersive liquid–liquid microextraction systems [22].

Although polymers present some limitations in polarity range and high viscosity [23], their unique characteristics, such as surface activity, micellization, and reversible thermorheological phase behavior in water, make polymer-based ATPSs draw increasing attention to extracting and purifying natural components [24]. Several natural products, including curcumin from *Curcuma longa* [25], genipin from genipap [26], and polysaccharides from *Camellia oleifera* [27], had been extracted by polymer/salt, polymer/ionic liquids, and thermal-responsive polymer/DESs ATPSs, respectively. As one of the well-known ABA triblock copolymers, Pluronic^®^ is composed of hydrophilic block poly (ethylene oxide) (PEO) and hydrophobic block poly (propylene oxide) (PPO). In recent years, scholars have accumulated rich experience in their colloid-chemical behavior in aqueous solutions, including colloid stabilization, micelles, and liquid crystalline phases [24]. Moreover, Pluronic^®^ could form novel ATPSs together with DESs for the extraction and separation of Nar and rutin [28], curcumin [29], and penicillin G [30]. However, the research on the design and application of a non-aqueous biphasic extraction system by using Pluronic^®^ and DESs has not been reported so far.

Ethanol is an easily prepared, non-toxic, human health-safe, and renewable solvent. The use of ethanol in the extraction process is sustainable and environmentally friendly, and it has been widely used in the food and pharmaceutical industries [23]. Therefore, the development of a novel ethanolic two-phase system (ETPS) capable of extracting and separating CFs comprising triblock copolymers, DESs, and ethanol is relevant. 

This study aimed to explore a newly reusable liquid–liquid extraction system for the efficient enrichment and separation of natural CFs. Thirteen hydrophilic DESs were synthesized using different quaternary ammonium salts and proline as HBAs, together with polyols and oxalic acid as HBDs. Moreover, a series of novel ETPSs composed of DESs and Pluronic^®^ L 64 (PL 64, Figure 1) were constructed and applied to investigate the distribution behavior of Nar and Neo standards. The experimental parameters such as the mass ratio of DESs and PL 64, DES concentration, PL 64 concentration, molar ratio of HBAs and HBDs, temperature, phosphate buffer solution, and ethanolic pH were performed. The application of the optimal ETPS for the extraction and separation of Neo and Nar from pomelo peel extract was studied. Moreover, the recovery of target CFs from DESs was investigated. Thereby, it is expected that the proposed ETPS would provide a simpler experimental operation for CFs separation and accelerate the development and application of new liquid–liquid extraction systems.

## 2. Materials and Methods

### 2.1. Materials and Reagents 

Pomelo was purchased from Lvhang Fruit Co., Ltd. (Zhangjiajie, China). PL 64 (PEO_13_-PPO_30_-PEO_13_, average molecular weight ≈ 2900 g/mol), choline chloride (ChCl, ≥99%), betaine (Bet, ≥98%), xylitol (Xyl, ≥99%), *L*-(-)-proline (Pro, ≥99%), tetraethylammonium chloride (N_2222_Cl, ≥98%), tetrabutylammonium chloride (N_4444_Cl, ≥95%), tetraethylammonium bromide (N_2222_Br, ≥98%), tetrapropylammonium bromide (N_3333_Br, ≥98%), tetrabutylammonium bromide (N_4444_Br, ≥99%), and *D*-(+)-maltose monohydrate (Mal, ≥99%) were provided by Adamas Reagent Co., Ltd. (Shanghai, China). Glycerol (Gly, ≥99%), *D*-(+)-glucose (Glu, ≥99.7%), oxalic acid dihydrate (OA, ≥99%), ethanol (≥99.7%), citric acid (CA, ≥99.5%), phosphoric acid (PA, ≥85%), ethyl acetate (EA, ≥99.5%), potassium phosphate dibasic trihydrate (K_2_HPO_4_·3H_2_O, ≥99%) were purchased from Greagent Reagent Co., Ltd. (Shanghai, China). Neo (99.69%) and Nar (99.88%) were purchased from RefMedic Biotech Co., Ltd. (Chengdu, China). Other reagents were analytically pure and used without further treatment.

### 2.2. Preparation of DESs

DESs were prepared by means reported in the literature [16,17], which are shown in Table 1. Briefly, HBAs and HBDs (Appendix A) at a certain molar ratio were accurately weighed into a round-bottle flask and subjected to magnetic stirring at 80–100 °C for 1–2 h until a homogeneous liquid formed.

### 2.3. Determination of the Phase Diagrams

The binodal curve was constructed by a cloud-point titration method [31]. In brief, experimental systems with different weight fractions of N_4444_Br and PL 64 were prepared and kept the final weight constant at 1 g in a glass tube. Then the mixtures were shaken vigorously for 3 min in a QL-866 vortex Mixer. Ethanol was added to this tube using a microsyringe until the mixtures turned clear. The corresponding quality was recorded. The data were fitted by employing an empirical Merchuk Equation (1) [32].
(1)$${Y=\mathrm{Aexp}}^{{[(\mathrm{BX}}^{0.5})\;-{\;(\mathrm{CX}}^{3})]}$$
where Y and X are the mass fractions of PL 64 and DESs, respectively. A, B, and C are the fitting parameters.

### 2.4. Extraction of Nar and Neo Using DESs/PL 64 ETPSs

A certain amount of DESs, PL 64, and ethanol was added to a glass centrifuge tube. Then 1 mg of Nar and Neo were weighed into this tube. Afterward, the mixed solution was well blended on a QL-866 vortex Mixer for about 3 min and placed in a constant thermostat bath. After equilibrium, the mixed solution was centrifuged with a TDZ5-WS centrifuge (Changsha PingFan Instrument Co., Ltd., Changsha, China) for 10 min at 2000 rpm. The biphasic phase volumes were recorded. The concentrations of Nar and Neo were determined by HPLC analysis. Each experiment was performed in triplicate runs. Three parameters of the extraction efficiency (*E*), partition coefficient (*K*, Appendix A), and selectivity (*S*) were defined in Equations (2)–(4), respectively.
(2)$$\;E\left(\%\right)=\frac{C_{\mathrm{DESs}}{\times V}_{\mathrm{DESs}}}{C_{\mathrm{DESs}}{\times V}_{\mathrm{DESs}}{+C}_{\mathrm{PL}\;64}{\times V}_{\mathrm{PL}\;64}}\times 100\%$$
(3)$$K=\frac{C_{\mathrm{DESs}}}{C_{\mathrm{PL}\;64}}\;\;$$
(4)$$S=\frac{K_{\mathrm{Neo}}}{K_{\mathrm{Nar}}}\;\;$$
where *C*_DESs_ and *C*_PL 64_ are the Nar and Neo concentrations in DESs-rich and PL 64-rich phases, respectively. *V*_DESs_ and *V*_PL 64_ are the volumes of DESs-rich and PL 64-rich phases, respectively.

### 2.5. Recovery of Nar and Neo from DESs

D101 macroporous resin of 10 g was soaked in ethanol and filled into a glass column (15 mm × 50 cm) and performed for the recovery of DESs and target flavonoids. The bed volume (BV) of this resin was 20 mL. The DESs-rich phase of 2.2 mL was obtained under the pomelo peel extract separated by DESs/PL 64 ETPS was added into this resin column, successively eluted with 120 mL of water and 95% aqueous ethanol at a flow rate of 3 BV/h, to obtain DESs and CFs eluents, respectively. The collected water and ethanol eluents were diluted and analyzed by HPLC as described in Appendix A and Section 2.6. The recovery yields (*RYs*) of DESs and target CFs were calculated. These experiments were performed in triplicate.

### 2.6. HPLC Analysis

Nar and Neo concentrations were determined by an Agilent 1260 HPLC system (Agilent, San Jose, CA, USA), which was equipped with a G1315D DAD detector. An Agilent C18 column (4.6 mm × 250 mm, i.d., 5 μm) was used for separating samples. Regarding the isocratic elution, a mobile phase comprising methanol and 0.1% phosphoric acid solution (45:55) was utilized. The analytical conditions of column temperature, flow rate, injection volume, and detection wavelength were maintained at 30 °C, 1.00 mL/min, 20 μL, and 283 nm, respectively. The analyses were performed in triplicate. The chromatograms and standard fitting curves are shown in Appendix A. 

## 3. Results and Discussion

### 3.1. The Phase Behavior of DESs/PL 64 ETPS

Data of the binodal curve for ETPS composed of DESs, PL 64, and ethanol was obtained. Figure 2 indicates that the liquid–liquid two-phase region of DESs/PL 64 ETPS is narrow, which is basically consistent with that of the reported polymer/ionic liquid ATPSs [31]. Data of the mass fraction of system components (Appendix A) were successfully fitted by using the Merchuk Eq., which were displayed as follows:$${Y=120\exp}^{\left[\left(-0{.1169X}^{0.5}\right)-\left(4{.5114\;\times \;10}^{-6}X^{3}\right)\right]}$$

An R^2^ of 0.9963 indicates that the data for this ETPS binodal fitting are sufficiently satisfactory [33].

### 3.2. Extracting of Nar and Neo in Different DESs/PL 64 ETPSs

#### 3.2.1. Effect of the DESs types 

Nar and Neo are easily soluble in strong polar organic solvents [8]. Eight hydrophilic DESs were synthesized using quaternary ammonium salts (ChCl, Bet, and N_2222_Cl) and Pro as HBAs, together with polyols (Mal, Glu, Xyl, and Gly) and OA as HBDs, to improve the solubility of Nar and Neo. A series of ETPSs composed of 30 wt% DESs, 35 wt% PL 64, and 35 wt% ethanol were constructed to evaluate the distribution behavior of Nar and Neo (Figure 3a). The PL 64 and DESs in these ETPSs are enriched in the top and bottom phases, respectively. Except for one ETPS containing ChCl-OA, Nar and Neo are mostly enriched in the DESs-rich phases. The *E*_Nar_ and *E*_Neo_ values varied at 45.57–87.18% and 48.22–88.49%, respectively. The standard deviations of *S* values are high in glycosyl-based DESs ETPSs because these DESs containing polyols of Mal, Glu, and Xyl exhibit a high-viscosity system, which affects the uniform distribution of target analytes. Moreover, the viscosity of ETPS comprising ChCl-Gly DES is low, and its displayed *S* value of 1.76 is higher than that of the others. Therefore, DESs comprising Gly and various quaternary ammonium salts were synthesized and further optimized to investigate their effects.

#### 3.2.2. Effect of the Alkyl Chain Length for DESs

Owing to most of the DESs/PL 64/ethanol mixed systems could not completely form liquid–liquid two-phase at a mass ratio of 30 wt%/35 wt%/35 wt%, the mass ratios for ETPSs were adjusted to 43 wt%/43 wt%/14 wt% for subsequent studies. Figure 3b indicates that the top phases of two ETPSs of N_2222_Cl-Gly/PL 64 and N_4444_Cl-Gly/PL 64 are DESs-rich phases, which are different from the others investigated because the low-density DESs are induced by the presence of Cl^–^; the *E*_Nar_ and *E*_Neo_ values decrease as the alkyl chain length of DESs increasing, probably because a steric hindrance effect of longer alkyl chain weakens the H-bond interactions of DESs/target flavonoids [34]. N_4444_Br-Gly/PL 64 ETPS exhibits a lower *E* values (*E*_Nar_ = 74.97%, *E*_Neo_ = 84.48%) and a higher *S* value (*S* = 1.70) than those of N_4444_Cl-Gly/PL 64 ETPS. This phenomenon might be that the electronegativity of Br^–^ in N_4444_Br-Gly was weaker than that of Cl^–^ in N_4444_Cl-Gly, thereby resulting in a weak ability of N_4444_Br-Gly to form H-bonds with Nar and Neo [34]. Therefore, N_4444_Br-Gly and PL 64 were used as the optimal phase compositions for subsequent studies.

### 3.3. Single-Factor Experiments

The distribution behavior of Nar and Neo in N_4444_Br-Gly/PL 64 ETPS was investigated by single-factor experiments. The influencing factors of the mass ratio of DESs and PL 64, DES concentration, PL 64 concentration, molar ratio of HBAs and HBDs, temperature, phosphate buffer solution, and ethanolic pH were performed.

#### 3.3.1. Effect of the Mass Ratio of DESs and PL 64 

The effect of the mass ratios of DESs and PL 64 at 50:34–15:79 (wt%:wt%) was investigated. Figure 4a indicates that the *E*_Nar_ decreases with the decrease in mass ratios; while there is no obvious variation for the *E*_Neo_ values in the range of 50:34–22:70, and *E*_Neo_ decreases significantly to 47.25% when the mass ratio is 15:79. The reason for this phenomenon may be that viscosity of the PL 64-rich (top) phase increases as the decreasing of diluent ethanol in the top phase, promoted the migration of target CFs to the top phase driven by stronger micelle action. The *S* value reaches the maximum of 2.88 at 29:61. Thereby, the mass ratio of 29:61 was chosen.

#### 3.3.2. Effect of the Concentration of DESs 

N_4444_Br-Gly DES was investigated at concentrations of 27–35 wt% when the PL 64 concentration was 61 wt% to evaluate their effects. Figure 4b indicates that the *E*_Nar_ values gradually increase from 51.62% to 89.19% within the investigation concentration range; the *E*_Neo_ increases from 65.24% to 86.09% when the DES concentration increases from 27 wt% to 29 wt%; however, there is no significant change as the concentration further increases. Moreover, ETPSs with low DES concentration exhibit a large difference in *E*_Nar_ and *E*_Neo_ values in the same phase, which are more efficient for the distribution of Nar and Neo in the two phases. Thus, 29 wt% of DESs concentration was chosen for further studies.

#### 3.3.3. Effect of the Concentration of PL 64 

The effect of PL 64 concentration in this ETPS was investigated in a range of 59–67 wt% when the DES concentration was 29 wt%. The results in Figure 4c show that Nar and Neo are mainly enriched in the PL 64-rich phase when the concentration of PL 64 is 59 wt% (*E*_Nar_ = 43.79%, *E*_Neo_ = 48.94%). The *E*_Nar_ and *E*_Neo_ values increase as the further increase in PL 64 concentration; *E*_Nar_ values gradually increase, and *E*_Neo_ increases first and then decreases, reaching a maximum value of 86.09% at the PL 64 concentration of 61 wt%. Therefore, the optimal PL 64 concentration of 61 wt% was used.

#### 3.3.4. Effect of the Molar Ratio of HBAs and HBDs 

The molar ratio of HBAs and HBDs is very crucial for the physiochemical properties and extraction abilities of DESs [35]. The effect of N_4444_Br and Gly at the mole ratios of 1:3–1:7 was evaluated. The phase volume ratios for different N_4444_Br-Gly/PL 64 ETPSs are about 2.6:0.8 at the molar ratios of 1:3–1:7, which has not changed (Appendix A). Figure 4d indicates that the variation ranges of *E*_Nar_ and *E*_Neo_ values are 59.42–68.32% and 70.95–86.09%, respectively, and their changing trends are not obvious; the *S* values first decrease and subsequently increase in the investigation region. Unfortunately, N_4444_Br-Gly DESs formed by different molar ratios have no significant effect on the selective distribution of Nar and Neo. When the molar ratio is 1:3, the maximal *S* value is 2.88. Thereby, the molar ratio of N_4444_Br and Gly was fixed at 1:3.

#### 3.3.5. Effect of the Equilibrium Temperature 

The effect of temperature in the range of 15–55 °C was studied (Figure 4e). The variation trends of *E*_Nar_ and *E*_Neo_ values indicate that the temperature has not significantly influenced the distribution of target CFs. As the temperature increase, *E*_Nar_ values first decrease and subsequently increase; however, *E*_Neo_ values first increase and then decreases, and the minimal *E*_Nar_ of 68.32% and *E*_Neo_ of 86.09% are obtained at 25 °C, respectively. The *E* values of Nar and Neo in the same phase tend to be close and result in a decreasing trend for *S* values with the temperature increasing, indicating that high temperature is not conducive to the selective separation of Nar and Neo. This phenomenon might be that high temperature caused the biphasic solutions of ETPSs to exhibit low viscosity and high fluidity. Thus, the micelle effect of the PL 64-rich phase was weakened, while the mass transfer effect of the EDS-rich phase was enhanced, thereby resulting in the enrichment of Nar and Neo in the same phase. Therefore, 25 °C of temperature was selected.

#### 3.3.6. Effect of the Phosphate Buffer Solutions

Phosphate buffer solutions with different pH (Appendix A) were prepared to investigate their effect on the distribution of Nar and Neo. Interestingly, excess buffer solution added to these ETPSs can destroy the formation of biphasic phases. Hence, only 20 mg buffer solution was added to these ETPSs for experiments. Figure 5a indicates the trends of *E*_Nar_ and *E*_Neo_ values first decrease and subsequently increase as the pH of buffer solutions increases. In addition, the minimum *E* values (*E*_Nar_ = 39.88%, *E*_Neo_ = 42.97%) were obtained at pH 5, indicating that target CFs tend to distribute into the PL 64-rich (top) phase. The ETPSs with buffer solutions do not exhibit an obvious increase in *S* value. Therefore, the ETPS without adding buffer solutions was chosen.

#### 3.3.7. Effect of the Ethanolic pH 

Generally, the solubility of Nar and Neo decreases in acid solution and increases in alkaline solution. Therefore, CA and NaOH were used to adjust the pH values of ethanol at 1.24–13.78 (Appendix A) to investigate their effect on the extraction of Nar and Neo. The *E*_Nar_ and *E*_Neo_ values shown in Figure 5b exhibit an irregular variation trend. Especially in the ETPSs with adding strong alkaline ethanol of pH 13.78, Nar and Neo are still mostly enriched in the DESs-rich (bottom) phase. Since ethanol without pH adjustment showed a higher *S* value, absolute ethanol was used in the pomelo peel extract separation.

### 3.4. Extraction of Nar and Neo from the Pomelo Peel

The optimal N_4444_Br-Gly/PL 64 ETPSs were applied for the extraction of Nar and Neo from the pomelo peel extract. Due to the low concentration of Neo in the pomelo peel extract, 0.2 mg pure Neo was added to each investigation ETPS. Compared with the standard chemical experiments, the *K* and *E* values in the actual sample experiments increase (Table 2). Interestingly, the *K*_Neo_ and *E*_Neo_ values increase more obviously than those of Nar; the *S* value decreases from 2.88 to 1.17. The pollutants from the pomelo peel ethanol extract have little effect on the distribution of Nar between the top and bottom phases but have a great effect on the distribution of Neo, indicating that the impurities dissolved in the PL 64-rich (top) phase promote additional Neo to migrate into the top phase.

### 3.5. Recovery of Nar and Neo from DESs

Due to the strong interaction between Nar/Neo and N_4444_Br-Gly, recycling them is a challenging task. Two methods of EA back-extraction and D101 resin adsorption were used to recover DESs, Nar, and Neo. The results showed that the *RYs* of Nar and Neo were 1.27% and 1.48%, respectively, in 5 times hot EA back-extraction (6.6 mL each time); the *RYs* of DESs, Nar, and Neo reached 78.12 ± 1.30%, 66.61 ± 2.14%, and 68.03 ± 1.61%, respectively, adsorbed by D101 macroporous resin. It can be seen that D101 resin adsorption can effectively recover target CFs and DESs based on different molecular sizes and provide a potential for the industrial utilization of this ETPS method.

## 4. Conclusions

In this work, a novel and environmentally friendly ETPS based on PL 64 and DESs was applied for the extraction and separation of Nar and Neo. The results revealed that the alkyl chain length and anion in quaternary ammonium salts significantly influenced the distribution of Nar and Neo; the biphasic region of N_4444_Br-Gly, PL 64, and ethanol was narrow. The extraction conditions for the N_4444_Br-Gly/PL 64 ETPSs were optimized via single-factor experiments. Moreover, the optimal extraction efficiency for Nar and Neo (*E*_Nar_ = 68.32%, *E*_Neo_ = 86.12%, and *S* = 2.88) was obtained under the following conditions: the mass ratio of DESs and PL 64 of 29:61, DES concentration of 29 wt%, PL 64 concentration of 61 wt%, molar ratio of HBAs and HBDs of 1:3, temperature of 25 °C, and without adding phosphate buffer solution. The *RYs* of DESs, Nar, and Neo were 78.12 ± 1.30%, 66.61 ± 2.14%, and 68.03 ± 1.61%, respectively, absorbed by a D101 macroporous resin column. This novel N_4444_Br-Gly/PL 64 ETPS was efficient for the extraction of Nar and Neo from pomelo peel and could be employed as a suitable strategy for green liquid–liquid extraction of natural products.

## Figures and Tables

**Figure 1 foods-11-02590-f001:**
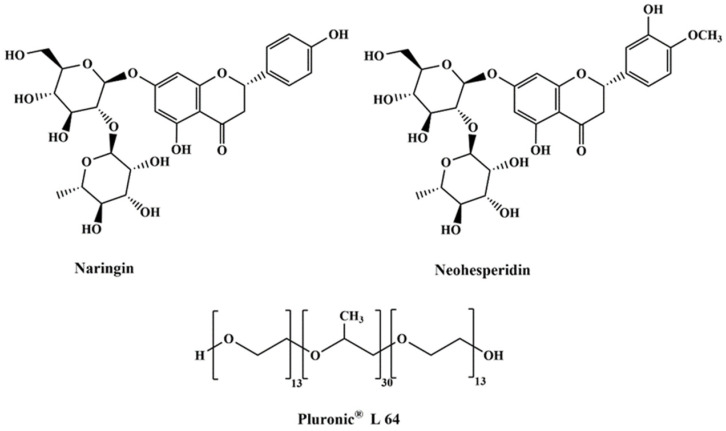
Chemical structures of Nar, Neo, and PL 64.

**Figure 2 foods-11-02590-f002:**
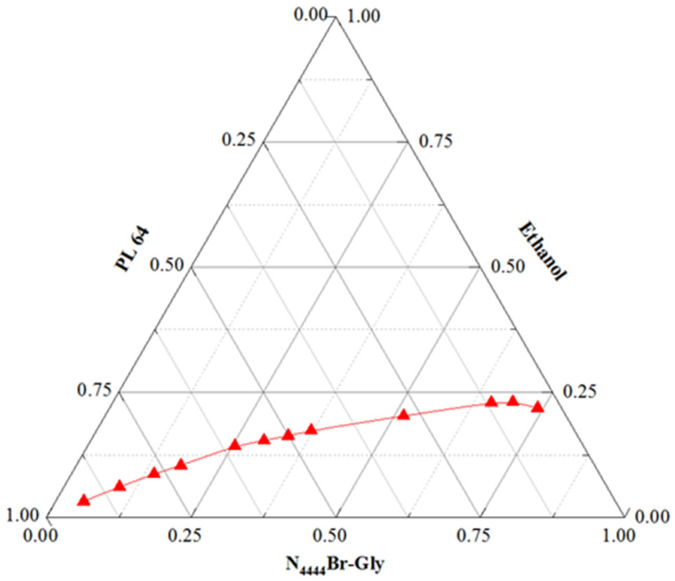
Phase diagram for system composed of N_4444_Br-Gly, PL 64, and ethanol at 25 °C and atmospheric pressure. Compositions are given as weight fractions. Red triangle symbols represent the experiment values.

**Figure 3 foods-11-02590-f003:**
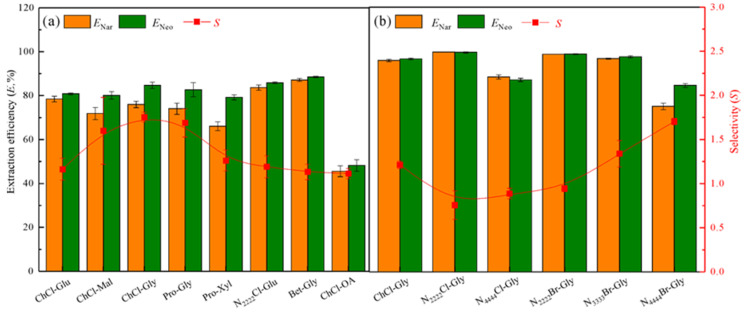
*E* and *S* values for Nar and Neo in different ETPSs. The extraction conditions were as follows: (**a**) DESs of 30 wt%, PL 64 of 35 wt%, temperature of 25 °C, (**b**) DESs of 43 wt%, PL 64 of 43 wt%, temperature of 25 °C.

**Figure 4 foods-11-02590-f004:**
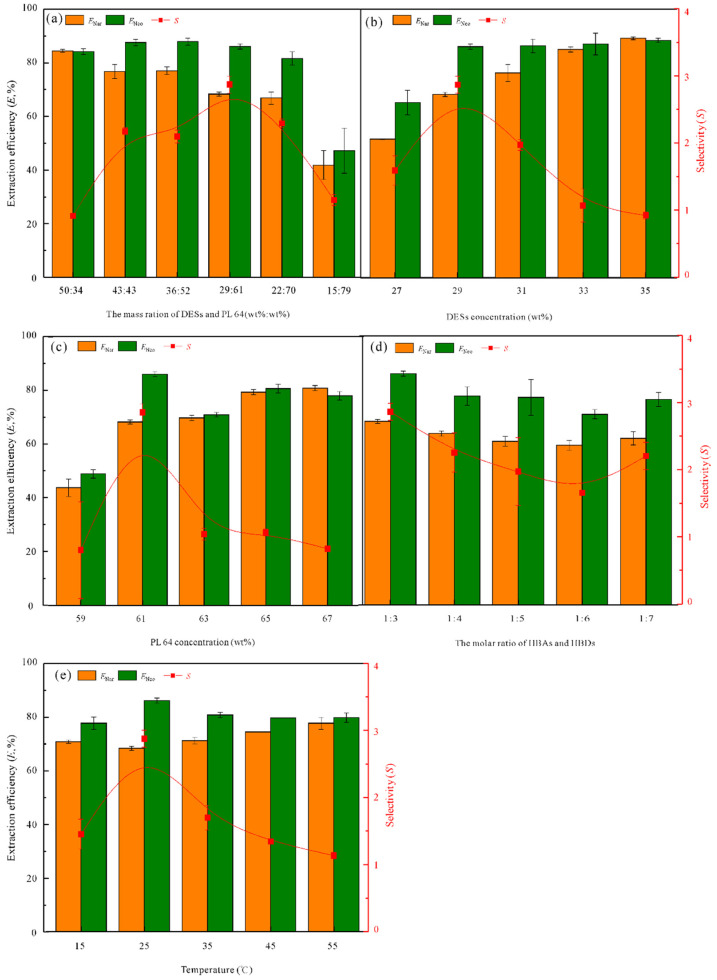
Effects of the (**a**) mass ratio of DESs and PL 64, (**b**) DES concentration, (**c**) PL 64 concentration, (**d**) molar ratio of HBAs and HBDs, and (**e**) equilibrium temperature on the *E* and *S* values. The extraction conditions were as follows: (**a**) temperature of 25 °C, (**b**) PL 64 of 61 wt%, temperature of 25 °C, (**c**) DESs of 29 wt%, temperature of 25 °C, (**d**) DESs of 29 wt%, PL 64 of 61 wt%, temperature of 25 °C, (**e**) DESs of 29 wt%, PL 64 of 61 wt%.

**Figure 5 foods-11-02590-f005:**
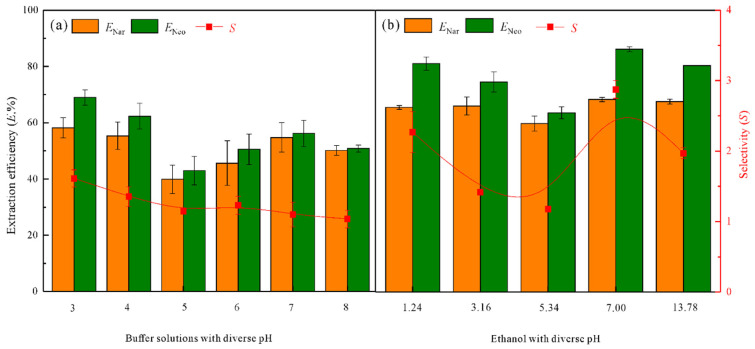
Effects of the (**a**) buffer solutions with diverse pH and (**b**) ethanol with diverse pH on the *E* and *S* values. The extraction conditions were as follows: DESs of 29 wt%, PL 64 of 61 wt%, temperature of 25 °C.

**Table 1 foods-11-02590-t001:** The abbreviations, molar ratios, pH, and densities of DESs used.

Abbreviations	HBAs	HBDs	Water	Mole Ratios	pH	Densities (g/cm^3^)
ChCl-Mal	Choline chloride	*D*-(+)-Maltose monohydrate	H_2_O	3:1:3	5.04	1.2387
ChCl-Glu	Choline chloride	*D*-(+)-Glucose	H_2_O	2:1:3	5.92	1.2206
ChCl-Gly	Choline chloride	Glycerol		1:2	7.45	1.1912
ChCl-OA	Choline chloride	Oxalic acid		1:1	1.00	1.2193
Bet-Gly	Betaine	Glycerol		1:2	8.73	1.2283
Pro-Gly	*L*-(-)-Proline	Glycerol		2:5	7.60	1.2757
Pro-Xyl	*L*-(-)-Proline	Xylitol	H_2_O	1:1:5	6.73	1.2648
N_2222_Cl-Glu	Tetraethylammonium chloride	*D*-(+)-Glucose	H_2_O	2:1:3	5.42	1.1321
N_2222_Cl-Gly	Tetraethylammonium chloride	Glycerol		1:1	7.60	1.0385
N_4444_Cl-Gly	Tetrabutylammonium chloride	Glycerol		1:2	5.11	1.0394
N_2222_Br-Gly	Tetraethlammonium bromide	Glycerol		1:2	10.14	1.2315
N_3333_Br-Gly	Tetrapropylammonium bromide	Glycerol		1:2	5.97	1.1650
N_4444_Br-Gly	Tetrabutylammonium bromide	Glycerol		1:3	6.63	1.1307
				1:4	6.80	1.1407
				1:5	6.18	1.1516
				1:6	6.46	1.1679
				1:7	5.81	1.1644

**Table 2 foods-11-02590-t002:** *K*, *E*, and *S* values in N_4444_Br-Gly/PL 64 ETPSs for the extraction of Nar and Neo from the standard and pomelo peel extract samples. The extract was prepared by adding 10 mL ethanol to 400 mg peel powders assisted by an ultrasonic cell pulverizer at 40 °C for 30 min.

Parameters	N_4444_Br-Gly/PL 64 ETPSs (29 wt%:61 wt%)	
	Standard Sample	Pomelo Peel Extract
*K* _Nar_	6.66 ± 0.41	5.90 ± 0.55
*E* _Nar_	31.68 ± 0.77%	34.92 ± 3.11%
*K* _Neo_	19.13 ± 0.90	6.87 ± 0.27
*E* _Neo_	13.91 ± 0.95%	31.48 ± 1.77%
*S*	2.88 ± 0.13	1.17 ± 0.06

## Data Availability

Data is contained within the article or Appendix A.

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
