# Peer review of "A Novel Ethanolic Two-Phase System Based on Deep Eutectic Solvents and Amphiphilic Copolymer for the Extraction of Neohesperidin and Naringin from the Pomelo Peel"

_foods, 2022, doi:10.3390/foods11172590_

Round 1

Reviewer 1 Report

This work is focussed on the development of a novel ethanolic two-phase system (ETPS) capable of extracting and separating the Citrus flavonoids (CFs) naringin (Nar) and neohesperidin (Neo) from the pomelo peel using triblock copolymers, deep eutectic solvents (DESs), and ethanol. The authors studied the effects of the (a) the mass ratio of DESs and Pluronic®L-64 (PL 64), (b) DESs concentration, (c) PL 64 concentration, (d) the molar ratio of hydrogen bond acceptors (HBAs) and hydrogen bond donors (HBDs), and (e) equilibrium temperature on the extraction efficiency (E) and separation selectivity (S). Their results showed that the alkyl chain length and anion in quaternary ammonium salt significantly influenced the distribution of Nar and Neo; the biphasic region of the optimal DES N4444Br-Gly, PL 64, and ethanol was narrow. The optimal extraction efficiency for Nar and Neo (ENar = 68.32%, ENeo = 86.12%, and S = 2.88), was obtained under the following conditions: the mass ratio of DESs and PL 64 of 29:61, DESs concentration of 29 wt%, PL 64 concentration of 61 wt%, temperature of 25 °C, and without adding phosphate buffer solution. They concluded that the N4444Br-Gly/PL 64 ETPS was efficient to extract Nar and Neo from the pomelo peel and could be employed as a good strategy for green liquid-liquid extraction of natural products.

I have some questions and comments

1) Do the authors have any idea about the interactions between the DESs, Pluronic®L-64, ethanol and the Citrus flavonoids?.
2) What will be the supramolecular structure?.
3) The difference in the chemical structures of Nar and Neo appears in the phenolic ring (OCH3 and OH position), how can this difference influence the separation process of those flavonoides?.
4) The scientific community has been interested in finding new methods for the extraction of natural products, what is the ratio yields/cost with this methodology?.
5) On page 5, line 165, "...and ethanol was obtained..." should read "...and ethanol was obtained..."

The manuscript is interesting, well written and easy to read.

Reviewer 2 Report

A brief summary:

The topic reveals innovative ways to green extraction technique of DES what makes the manuscript interesting. All parts of the study are described in understandable and concise way though there are some minor remarks that need to be taken into account.  

 General concept comments:

·         The Abstract is too long. It should include 200 words maximum. Please shorten the abstract according to the Instructions for authors.

·         Introduction is well written and needs some minor changes. Basic facts like DES definition are not needed here (see the specific comments below).

·         In the end of the Introduction, please separate the aim of the study in one or two sentences, avoiding too much description which should be transferred in the Methods and Materials.

·         Suggestion: In the Keywords separate „extraction and separation“ and „neohesperidin and naringin“ as two separate keywords  

·         Suggestion: Always use the same number of decimal places in the article.

·         Almost half of the references are too old. References older than 5 years could be included if there is no proper replacement. Where possible, please find and replace old literature with newer one.

·         Results and discussion present mostly the results. It is recommended to compare the results with other studies to a greater extent and present a discussion based on these findings.

·         Please check the text again for some typos (spaces between words or paragraphs or similar).

·         The English language should be checked with the professional for better understanding.

 Specific comments:

·         Line 17-18: Please rewrite the sentence. It can not start with „And“.

·         Line 31: Please exclude the reference number after the first sentence. It is listed after the second sentence.

·         Line 36-38: Please rewrite the sentence to make it more clearer.

·         Line 38: Please exclude the reference number after „antioxidant“. It is listed after „anti-inflammatory“ which includes both terms.

·         Line 50-51: Please rewrite the sentence. It can not start with „And“.

·         Line 52-54: Please include a reference.

·         Line 56: This sentence is unnecessary.

·         Line 62: Please change „designable“ to „designability“

·         Line 120: Please  remove the word „Then“ and rewrite the sentence.

·         Line 131: In Table 1. check Density units

·         Line 158: Please include term „isocratic mobile phase“ if the used mobile phase was isocratic.

·         Line 152-162: In the paragraph about HPLC analysis, please include how many times was done the analysis, for example: The analyses were done in triplicate.

Reviewer 3 Report

The authors of MS titeled "A novel ethanolic two-phase system based on deep eutectic sol- 2 vents and amphiphilic copolymer for the extraction of neohe- 3 speridin and naringin from the pomelo peel" were able to construst an effective method of separtion of 2 flavonoids close in their chemical structure to be isolated effectively from peels extract using green safe solvents. 

1. The question is why specifically the authors choose the isolation of these specific flavonoids neohespiridin and naringenin ? what is the rational of isolation these flavonoids from peels extract? 

2. I checked the pricing of these flavonoids online and I found they are not expensive especially naringenin so why the authors did not try for another flavonoids more expensive ? "from a commercial point of view"

3. Is the described green extraction method applicable for another flavonoids or should I try till reaching effective parameters for maximum flavonoid isolation ? I mean what are the criteria of  the paramters tested in the MS as per flavonoids chemical structure ?

4. will this green method of extraction be selective as there are other flavonoids present in the citrus extract ? I can see in line number 134, the authors added 1 mg of each flavonoid ? why did you add this 1 mg ?
